# Submaximal Elastic Resistance Band Tests to Estimate Upper and Lower Extremity Maximal Muscle Strength

**DOI:** 10.3390/ijerph18052749

**Published:** 2021-03-09

**Authors:** Bjarki T. Haraldsson, Christoffer H. Andersen, Katrine T. Erhardsen, Mette K. Zebis, Jéssica K. Micheletti, Carlos M. Pastre, Lars L. Andersen

**Affiliations:** 1Department of Physiotherapy, University College Copenhagen, DK-2200 Copenhagen, Denmark; christoffer@fitogsund.dk (C.H.A.); KTER@kp.dk (K.T.E.); mzeb@kp.dk (M.K.Z.); 2Department of Physiotherapy, São Paulo State University (UNESP), 305 Roberto Simonsen, Presidente Prudente, São Paulo 19060-900, Brazil; jessicamicheletti@hotmail.com (J.K.M.); pastre@fct.unesp.br (C.M.P.); 3National Research Centre for the Working Environment, DK-2100 Copenhagen, Denmark; LLA@nfa.dk

**Keywords:** muscle fatigue, resistance training, 1-repetition maximum, prediction

## Abstract

Muscle strength assessment is fundamental to track the progress of performance and prescribe correct exercise intensity. In field settings, simple tests are preferred. This study develops equations to estimate maximal muscle strength in upper- and lower-extremity muscles based on submaximal elastic resistance tests. Healthy adults (*n* = 26) performed a maximal test (1 RM) to validate the ability of the subsequent submaximal tests to determine maximal muscle strength, with elastic bands. Using a within-group repeated measures design, three submaximal tests of 40%, 60%, and 80% during (1) shoulder abduction, (2) shoulder external rotation, (3) hip adduction, and (4) prone knee flexion were performed. The association between number of repetitions and relative intensity was modeled with both 1st and 2nd order polynomials to determine the best predictive validity. For both upper-extremity tests, a strong linear association between repetitions and relative intensity was found (R^2^ = 0.97–1.00). By contrast, for the lower-extremity tests, the associations were fitted better with a 2nd order polynomial (R^2^ = 1.00). The results from the present study provide formulas for predicting maximal muscles strength based on submaximal resistance in four different muscles groups and show a muscle-group-specific association between repetitions and intensity.

## 1. Introduction

Assessing muscular strength is important for strength and conditioning coaches, as well as for physical therapists, e.g., for evaluating performance [1], for injury prevention and to predict risk of injury [2], for evaluating musculoskeletal dysfunction [3,4,5], and in rehabilitation [6]. Testing of muscle strength is also used at workplaces, where physical capacity is assessed in relation to work capacity [7], and further, to evaluate physical exercise programs, musculoskeletal disorders, and dysfunctions [8]. Muscle strength can be assessed in a number of ways, and is typically done so by using isokinetic or isometric testing devices, such as handheld-dynamometers [6,9,10], isokinetic dynamometers [1,2,4,11,12,13,14,15], conventional strength training machines or equipment, such as dumbbells or barbells and weight plates [7]. Of the above, isokinetic or isometric dynamometers have been considered as gold standard in assessing muscle strength for a number of years by researchers [16,17,18,19]. There is, however, a downside of using these devices; aside from often being rather large and stationary, they are expensive, require a certain degree of specific expertise, and can be rather time consuming.

Among clinicians, the use of portable, hand-held and easy-to-use devices has been on the rise, since they provide the opportunity to assess muscle strength in different settings, e.g., at the workplace, in the clinic, or on the field. However, the outcomes of these devices are influenced by the clinicians experience and technique [20,21,22,23]. Thus, a need exists for developing valid and reliable methods for evaluating muscle strength with portable and easy-to-use devices. One option is the elastic resistance band, which has a high validity and reliability for testing maximal muscle strength when compared with the gold standard of muscle strength evaluation [24]. Thus, since physical therapists and clinicians already have adapted this device into their practice, it would be practical to use these for muscle strength assessment and in the subsequent training, not only in the clinic but also in the workplace or at home, as they are both portable and ready-to-use. In addition, if provided with information regarding number of repetitions for specific training intensities, these professionals would be better able to secure accurate training progression.

With the high joint and muscle loads of maximal muscle strength testing, people who suffer from musculoskeletal pain may find it difficult or uncomfortable to perform such a test due to pain and/or discomfort, and they might, therefore, not reach true maximal strength values/levels. Therefore, estimating maximal strength through a submaximal test using elastic resistance bands is a good alternative, and might be preferable in many settings, especially where high levels of musculoskeletal pain are present. As muscle strength rapidly increases with resistance training in untrained individuals, a precise testing is not always necessary.

Therefore, this study aimed to develop equations to estimate maximal muscle strength in upper- and lower-extremity muscles based on submaximal elastic resistance tests and, in addition, to provide reference values regarding the required number of repetitions to reach certain intensity levels without having to perform a maximal muscle strength test.

## 2. Materials and Methods

### 2.1. Experimental Approach to the Problem

Using cross-sectional within-group repeated measures design, the study participants performed a maximal strength test (1 RM) and subsequently three submaximal tests at 40%, 60%, and 80% of this 1 RM during (1) shoulder abduction, (2) shoulder external rotation, (3) hip adduction, and (4) prone knee flexion, on separate occasions (see Figure 1). This design allows the determination of the association between number of repetitions and relative intensity, using both 1st and 2nd order polynomials to determine the best predictive validity for each of the different exercises.

### 2.2. Subjects

Twenty-six healthy men and women gave written consent to participate in the study (Table 1). The participants were recruited through personal contact and information-based introductions in the Department of Physiotherapy at University College Copenhagen, between October 2018 and May 2019. Inclusion criteria was an age between 18 and 67. Exclusion criteria were known cardiovascular diseases, chronic use of anti-inflammatory drugs, active inflammatory processes, and any musculotendinous or joint injury in shoulder, groin, or knee 6 months prior to the data collection. All participants received verbal and written information about the purpose and content of the study and signed an informed consent to participate. This conformed the Declaration of Helsinki and was approved by the Local Ethical Committee (H-3-2010-062).

### 2.3. Procedures

The study consisted of two sessions performed with on average 7 days apart (range 4–8). A session included four different exercises, two for the upper body and two for the lower body: (I) shoulder abduction (lateral raises), (II) shoulder external rotation, (III) hip adduction, and (IV) prone knee flexion (see Figure 2, Figure 3, Figure 4 and Figure 5). The exercises were selected based on being commonly used in clinical practice and being simple single joint movements targeting muscles and/or muscle dysfunction related to neck/shoulder pain (I), glenohumeral pain (II), groin injuries (III), and knee injuries (IV).

During the first session, anthropometric information was gathered, careful instructions of the exercises were given, and the participants were familiarized with the testing procedure. After warmup, consisting of a general whole-body warmup and few repetitions at low resistance for each exercise, the participants established their 1 RM resistance with elastic bands, as described in two previous studies [24,25]. Subsequently, they performed a submaximal test with elastic bands representing the nearest value of 60% of the 1RM level for all four exercises, respectively. The second session consisted of warmup and submaximal testing at the nearest value of 80% and 40% resistance level of the 1 RM, in that order, for all four exercises. Resistance levels for each exercise were separated by at least 30 min.

The elastic bands (TheraBand^®^ CLXTM, Performance Health, Akron, OH, USA) used in this study were of a standard length of 1.80 m, with resistance ranging from very low to very high (yellow, red, green, blue, black, silver, gold) (Figure 1). Prior to the onset of data collection, the elastic bands were stretched between 50–100 times [24], minimizing the possible changes in the mechanical properties of the elastic band during the initial stretching cycles. The same sets of elastic bands for each exercise were used throughout the study. The TheraBand^®^ CLXTM elastic bands have consecutive loops, and the loops at each end functioned either as handles or to secure the elastic band to wall-mounted stall bars, depending on which exercise was being performed. 

#### 2.3.1. 1 RM Tests

For all the exercises, the participants started with the elastic band with the least resistance (yellow CLX band–Figure 1) and worked their way up the resistance levels with approximately 1-min interval between levels, until they were unable to perform the exercise. The end position of each exercise consisted of 1-s static position. When failure occurred, one more attempt on the same level was made, with at least 1-min rest. Failure was determined when the end position could not be reached and/or if the l-second static end position could not be maintained. The pivot point in the knee flexion exercise, the shoulder external rotation and the hip adduction exercises were either at 100% or 200% of elastic band length (340 cm or 510 cm from wall mount, respectively), depending on the strength level of the participants. If the highest level (level 18) was reached at 100% length, the test continued at 200% with the same procedure as at 100%, starting with the elastic band with the least resistance. The validity and reliability of elastic resistance bands in measuring maximal muscle strength has been determined against isokinetic dynamometers in previous studies with good results [26,27] (also see in Discussion). Similar assessments were not performed in this study.

#### 2.3.2. Submaximal Tests

The submaximal tests consisted of performing the maximum possible number of repetitions until exhaustion. The participants were instructed to maintain a certain rhythm of repetitions (1.5–2 s in the concentric phase and 1–1.5 s in the eccentric phase) during the submaximal tests, to avoid breaks between each repetition. The requirements for a failed repetition were met when the exercise form was not maintained (compensatory movements of other body segments), and/or if the participant was not able to reach the required range of motion of the movement (the required range of motion was visually assessed by the test leader; also see description of each exercise further down), and if there was a break of 2–3 s between repetitions. The order of exercises performed was not fixed; however, the participants alternated between upper body and lower body exercises to avoid fatigue. Each participant performed the exercises in the same order on days 1 and 2.

For the upper-body exercises, all 26 participants performed both tests (1 RM and 60% of 1 RM) on day 1, on day 2 twenty-five and twenty-four participants performed the 40% of 1 RM and 80% of 1 RM, respectively. For the lower-body exercises, 25 participants performed all tests on days 1 and 2. The pain the participants experienced in the specific exercises were not related to participation in the project, according to the participants own information about their history of pain and injuries.

### 2.4. Exercises

#### 2.4.1. Shoulder Abduction-Standing Bilateral Raises

In the starting position, the participants were instructed to stand with feet together in the middle of the elastic band and to grab each end loop with their left and right hand (Figure 2A), with arms alongside the body. The participants were instructed to perform a standing bilateral shoulder lateral raise to 90° shoulder abduction angle and held for 1 s (Figure 2B). Correct shoulder angle was insured with visual inspection by the test leader.

#### 2.4.2. Shoulder External Rotation

In the starting position, the participants were standing with the dominant upper arm straight alongside the body and the elbow flexed at 90° in front of the body (in an internal rotation), gripping the elastic band with one hand. The end position was when the lower arm was perpendicular to the body (90° external rotation) (Figure 3A,B).

#### 2.4.3. Hip Adduction

In the starting position, the participants were in a supine position parallel to the wall, with feet apart and the elastic band around the ankle of the dominant leg. The end position was when the dominant leg was touching the foot of the person helping to secure the position (Figure 4A,B).

#### 2.4.4. Prone Knee Flexion

In the starting position, the participants were in a prone position with the elastic band around the ankle of the dominant leg. They were instructed to bring the lower leg up to 90° of knee flexion or perpendicular to the floor and held for 1 s (Figure 5A,B). The participants were instructed to keep movement of the pelvis to a minimum, by keeping the pelvis secure to the floor.

### 2.5. Sample Size Calculations

Based on previous literature, we expected a strong association between repetitions and intensity. For example, a previous study showed a clear association between repetitions and intensity during the shoulder abduction exercise using only 9 subjects [25]. However, for the generalizability of the results, we aimed to include a larger sample of at least 25 subjects.

### 2.6. Statistical Analysis

For statistical analysis, average values for intensity and repetitions, at both maximal and submaximal levels, were plotted on in an x-y coordinate system, and associations were fitted using first- and second-order polynomials. The criteria used to choose the best-fitted formula were following: the explained variance should be as high as possible (preferably > 90%) and the estimation of 1 RM should be as close to 100% as possible. The best-fitted formula, according to these criteria, was selected for further analyses.

## 3. Results

For both upper-extremity exercises, a strong linear association between repetitions and relative intensity was found (R^2^ = 0.97–1.00). By contrast, for the lower-extremity exercises, the associations were fitted better with a 2nd order polynomial (R^2^ = 1.00).

The equations for determining maximal muscle strength for each exercise are as follows:

Equation (1)—Shoulder Abduction (lateral raise) (explained variance 99.6%):*Intensity* = −1.2903 × *repetitions* + 101.53.(1)

Equation (2)—Shoulder External Rotation (explained variance 96.8%):*Intensity* = −1.1848 × *repetitions* + 101.42.(2)

Equation (3)—Hip Adduction (explained variance 100%):*Intensity* = 0.009 × *repetitions*^2^ − 1.495 × *repetitions* + 101.6.(3)

Equation (4)—Prone Knee Flexion (explained variance 100%):*Intensity* = 0.0109 × *repetitions*^2^ − 1.6496 × *repetitions* + 101.97.(4)

The number of repetitions (mean ± SD//SE) performed at 80%, 60%, and 40% for the four different exercises are presented in Table 2. Regarding maximal strength levels, the men were generally stronger than the women, but the relations between repetitions and intensity was not significantly different between the sexes.

Figure 6 shows the association between number of repetitions and exercise intensity, with the equations and the explained variance, for each of the four different exercises.

## 4. Discussion

The present study provides formulas for estimating maximal muscles strength based on submaximal endurance test with elastic band in four different exercises. Furthermore, the study shows that the association between repetitions and intensity is specific for each exercise in question. At lower relative intensities, the muscles of the lower body tested in the present study possessed in general a larger endurance capacity, i.e., more repetitions, than the muscles of upper body.

Both upper-extremity exercises showed a strong linear relationship between repetitions and relative intensity. This intensity-repetition association could be explained with a 1st polynomial equation. The lower extremity exercises also displayed a strong relationship, which could better be explained using a 2nd polynomial equations, as the lower extremity muscles demonstrated a larger endurance capacity than the upper body muscles at low relative intensities. Both muscle fiber type and metabolic factors may explain these differences.

It can be speculated that in complex exercises involving larger muscle groups, more repetitions can be performed at lower intensities due to slight alterations in exercise technique and muscle recruitment. Our data support this as participants performing the lateral raise and shoulder external rotation exercises -which involve the smaller muscles in the shoulder girdle- were able to complete fewer repetitions at lower intensities than in the hip adduction and leg curl exercises. This is also supported by Hoeger et al. [28] who found all subgroups of participants were able to complete significantly more repetitions in the leg press exercise-involving the hip and knee extensors-than in the lateral pulldown and bench press involving upper body muscles. Even fewer repetitions were completed in the arm curl and leg curl exercises involving mainly one muscle. Thus, to optimize training and rehabilitation, it is important to base RM on formulas specific to or close to the exercise used. Thus, considering the above, the number of repetitions that can be performed during elastic band submaximal exercises appears to be dependent on the muscle mass involved in the exercise. Besides, asynchronous recruitment may occur during submaximal exercises intensities [29], which will allow some muscle fibers to rest, while others are being used to maintain the desired production of force, and consequently could serve to delay fatigue. Considering that larger muscle groups have a greater absolute number of motor units available for recruitment during exercise, asynchronous recruitment in these cases may allow more overall rest for the muscle fibers, which again delays fatigue. Thus, it elucidates the importance of using different formulas for different muscles groups considering the exercise performed and the muscles used.

The metabolic capacity (oxidative and glycolytic) of the muscle seem to be dependent on the degree of capillarization around the muscle fiber, substrate availability, the mitochondrial content of the muscle and the activity of the enzymes involved in substrate utilization [30]. In addition, handling of various ions, either involved in the muscle work or ions produced by the working muscle, can have an impact on the performance [30,31]. And there is a large body of studies that show a greater oxidative capacity in the muscles in the lower extremities, than that in the upper extremity muscles [32,33]. Some of the contributing factors being a difference in substrate utilization at comparable workloads, with a peak in fat oxidation occurring at a lower intensity level in the arms than in the leg exercises [34,35]. Furthermore, irrespective of training status, the muscles of the upper extremities seem to be less oxidative and have a reduced capacity to extract oxygen from the circulation, compared to the muscles in the legs [35,36]. This corresponds well with the finding of Ahlborg and Jensen-Urstad, who observed a greater glycogen utilization in the arms than in the legs at corresponding exercise intensities, resulting in a larger lactate release (three-times as high) [37]. This difference in oxidative capacity between the upper- and lower-extremities is most likely due to an unequal training and activity status. Whereas, the same muscle or muscle group can respond and adapt to the need for either fine control, short intense effort, or prolonged activity, by changing their metabolism [38,39]. Although the literature is scarce, it appears that at the lower the intensity of the test, the more variability between both individuals [40] and exercises [28] is to be expected.

Use of elastic resistance bands in a training or rehabilitation settings has been increasing throughout the last 5–10 years, and the validity and reliability of this device to measure strength and endurance has been investigated in quite a few studies, either against the same or similar tests using dumbbells and isokinetic dynamometer, or an isometric dynamometer. Both methods are considered a gold standard method to measure muscular strength [16]. Manor et al. [41] measured the validity and reliability of the elbow flexion strength in older adults using elastic bands against dumbbells and an isokinetic dynamometer. They found the use of elastic resistance bands to significantly correlate to the use of dumbbells (r = 0.62, *p* < 0.01) and to the isokinetic dynamometer (r = 0.46, *p* < 0.01). Furthermore, they observed a high degree of test-retest reliability using elastic band, with Intraclass Correlation Coefficient (ICC) values of 0.89. Anderson et al. [24] measured the validity and reliability of elastic resistance bands for measuring shoulder muscle strength against an isometric dynamometer during a maximal voluntary contraction. For concurrent validity, the ICC value was 0.96 and the intratester reliability had an ICC value of 0.98. Both values considered to be of excellent validity and reliability.

Guex et al. [26] determined the validity and reliability of maximal strength assessments of knee flexors and knee extensors using elastic resistance bands. The validity was determined against an isokinetic dynamometer, and it was found to be very high for both knee flexors and extensors (r = 0.93 for both). The relative reliability for knee flexors and extensors was also found to be very high, with ICC values of 0.98 and 0.99, respectively. Nyberg et al. [27] investigated the validity of knee extensor strength measurements in older adults, using elastic resistance bands and a isokinetic dynamometer. The validity analysis showed a good agreement between the two methods, with overall ICC values of 0.88, and ICC values for men and women of 0.80 and 0.67, respectively.

As this study only tested healthy subjects, these results cannot be directly extrapolated to an unhealthy population, a different age population (elderly), people with muscular pain, or other muscles groups than tested here. Doing so should at least be done with caution. This study, however, tested both men and women, with age ranges from 20–43 years using two different exercises for both the upper and lower body extremities. Further studies should consider determine the relationship between resistance and repetitions in specific populations. The tests provided here are easy to expand on and test in other populations, and other research teams may provide updated information in other populations than healthy individuals.

The present results have several practical applications. First, performing maximal muscle strength tests (1 RM) is not always feasible e.g., due to safety aspects. The submaximal tests provided here—for each specific body part—can provide a good estimate of maximal muscle strength. Second, training intensity is a vital part of most strength training programs, e.g., training with 70% of maximal intensity. The intensity-repetitions charts and formulas provided here can be used to determine the number of repetitions maximum for each body part to reach a certain intensity without having to perform a maximal strength test.

## 5. Conclusions

The present study provides formulas for predicting maximal muscle strength based on submaximal resistance in four different exercises and shows that the association between repetitions and intensity is specific for each of the exercises in question.

## 6. Practical Applications

For a strength and condition coach, a clinician, or a physical therapist, assessment of muscle strength is often an important part of the physical examination. Strength assessment is performed in various ways with different downsides regarding the devices. In addition, manual muscle testing (MMT) can be used in rehabilitation as a part of the muscle strength assessment [42], as it is inexpensive and can be easily undertaken in different kinds of settings [43]; nonetheless, MMT has its limitations [44]. Since these groups of professionals often already use elastic resistance bands, it could be considered as an alternative for the muscle strength assessment. This could also facilitate the experience of a direct link between the assessment and the subsequent training/rehabilitation, from a patient point of view.

Considering that not all people are able to perform a maximal strength test due to pain or musculoskeletal dysfunctions, estimating the maximal strength from submaximal testing may be preferable. Elastic resistance bands seem to be a valid and a reliable tool for direct measurement of maximal muscular strength and endurance, which makes them an excellent alternative to the use of fitness equipment, or other expensive, large, and time-consuming apparatus, which often require specific expertise to operate, to either determine maximal strength and endurance or to simply train these qualities. This is pertinent for populations with functional limitations, impairments or pain, or any other population where use of traditional training equipment is not advised or cannot be performed due to pain or discomfort or accessibility (e.g., bedridden individuals). With the ease of use and low cost of elastic resistance bands, a regular training or assessment of muscular attributes could be used in a variety of settings, e.g., rehabilitation or prophylactic, not only in the clinic but also in the workplace or at home. Using the number of repetitions for each intensity provided by this study as a guideline, the professional can in a more reliable manner dose the training intensity and, thereby, secure a more stable progress throughout the training period.

## Figures and Tables

**Figure 1 ijerph-18-02749-f001:**
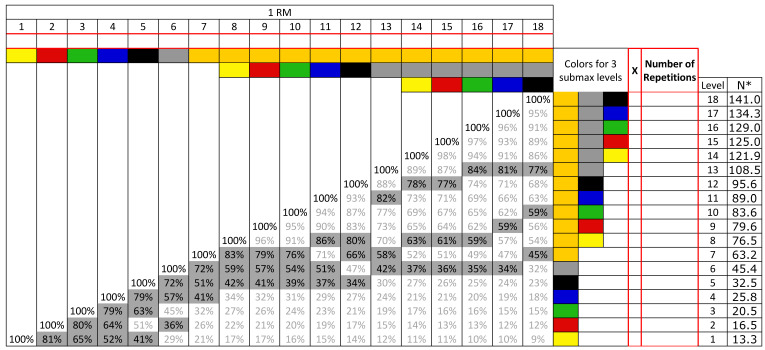
Color codes and levels of resistance of the elastic bands. One RM (one-repetition maximum) is the maximum load/resistance that an individual can lift just once with proper technique in the entire range of motion of the exercise. Top row: 1 repetition on each level until 1 RM is found. First column on the right (marked with red): Tick the intensities closest to 40, 60, and 80% (highlighted in gray in main table). Second column on the right (marked with red): Number of performed repetitions on each of the 3 levels to exhaustion. * Resistance in Newtons at 100% stretch length of the elastic band.

**Figure 2 ijerph-18-02749-f002:**
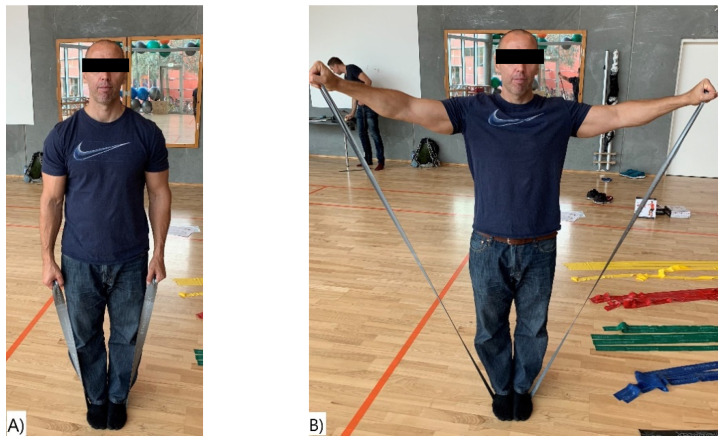
Shoulder Abduction. (**A**) Start position and (**B**) end position of the shoulder abduction exercise.

**Figure 3 ijerph-18-02749-f003:**
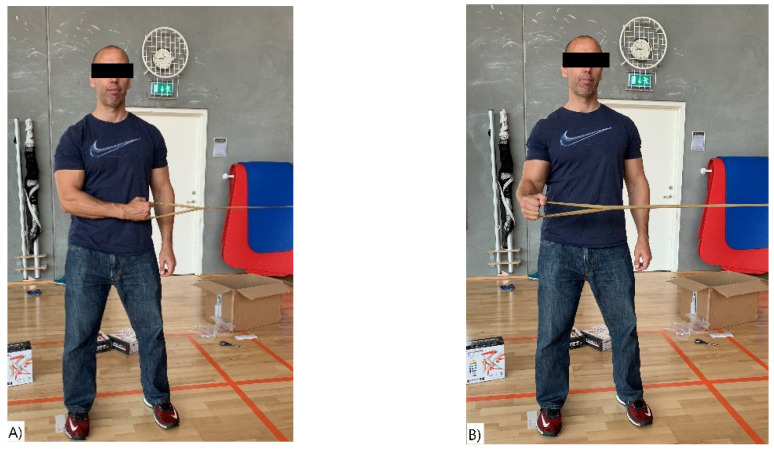
(**A**,**B**) Shoulder External Rotation. (**A**) Start position and (**B**) end position of the shoulder external rotation exercise.

**Figure 4 ijerph-18-02749-f004:**
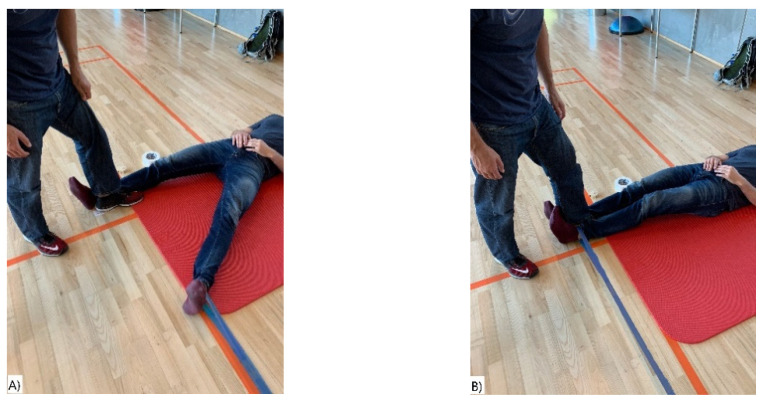
(**A**,**B**) Hip Adduction. (**A**) Start position and (**B**) end position of the hip adduction exercise.

**Figure 5 ijerph-18-02749-f005:**
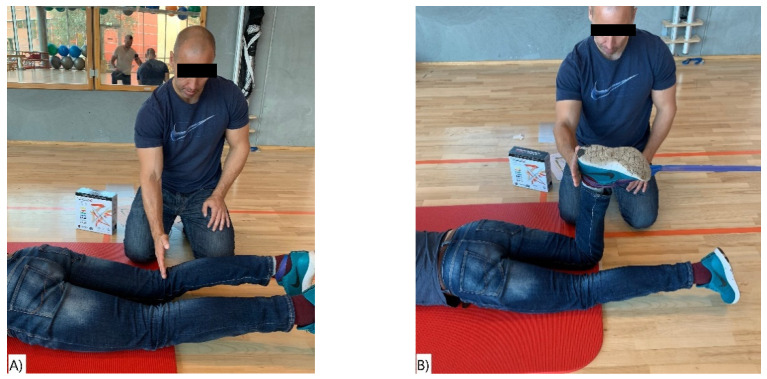
(**A**,**B**) Prone knee flexion. (**A**) Start position and (**B**) end position of the hamstring curls exercise.

**Figure 6 ijerph-18-02749-f006:**
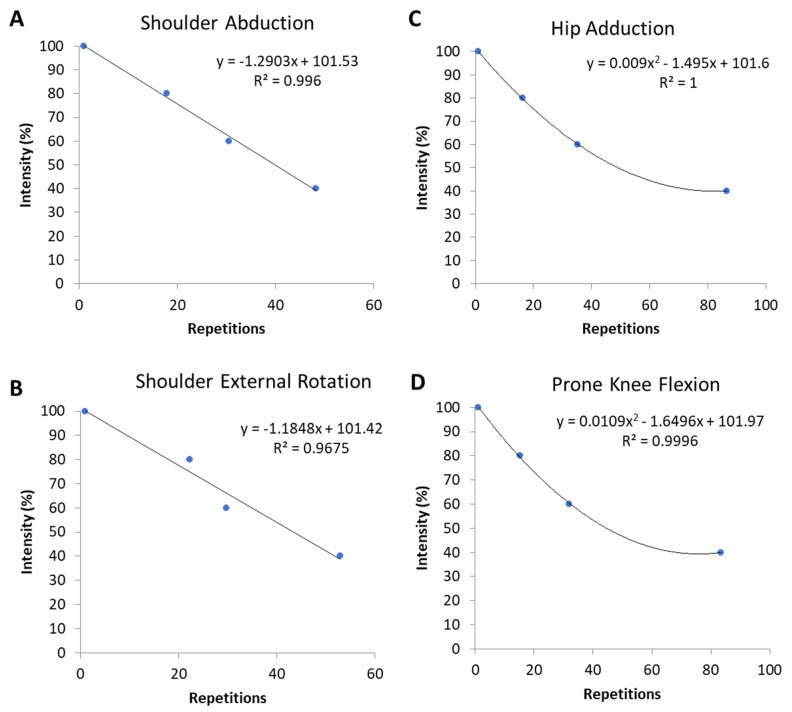
(**A**–**D**) The association between repetitions and intensity for upper and lower body exercises, fitted with 1st order polynomial ((**A**) Shoulder Abduction and (**B**) Shoulder External Rotation)) and 2nd order polynomial ((**C**) Hip Adduction and (**D**) Prone Knee Flexion)), respectively.

**Table 1 ijerph-18-02749-t001:** Participant characteristics.

Participants (men/women)	26 (11/15)
Age (y)	24.9 ± 6.15
Body mass (kg)	71.1 ± 12.15
Height (m)	1.75 ± 0.07
BMI (kg/m^2^)	23.0 ± 2.74
Smokers/non-smokers	1/25
Strength training sessions/week (average)	2.5 (range 0–8)

All parameters presented as mean ± SD. BMI = Body Mass Index.

**Table 2 ijerph-18-02749-t002:** Number of repetitions performed at 80%, 60%, and 40% of 1 RM for all four exercises.

Number of Repetitions Performed	Intensity			
	100% (SD)	80% (SD//SE)	60% (SD//SE)	40% (SD//SE)
Shoulder abduction	1 (n/a)	18 (8.7//1.8)	30.5 (17.8//3.5)	48.3 (20.5//4.1)
Shoulder external rotation	1 (n/a)	22.3 (7.1//1.5)	29.8 (9.4//1.8)	53.0 (14.2//2.8)
Hip adduction	1 (n/a)	16.2 (11.2//2.2)	35.2 (16.4//3.3)	86.7 (56.0//11.2)
Prone knee flexion	1 (n/a)	15.3 (8.7//1.7)	31.8 (14.0//2.8)	83.4 (63.3//12.7)

All results are presented as mean ± SD//SE.

## Data Availability

The data used to support the findings of this study are available from the corresponding author upon request.

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
