# Peer review of "Submaximal Elastic Resistance Band Tests to Estimate Upper and Lower Extremity Maximal Muscle Strength"

_ijerph, 2021, doi:10.3390/ijerph18052749_

Round 1

Reviewer 1 Report

The article has merit and practical use. The descriptions of the exercise are clear and the pictures help. There are other articles and organizations that have evaluated sub-maximal testing to predict 1 RM, a discussion of this would add to the article. The discussion talks more about the physiology than what this means for other clinicians. Please work on describing the “so what?” to the article. How can people use this information moving forward?

Line 109: Are there any references for why you stretched the bands 50-100 times? Is there a point where the bands loose tensile strength/elastic damage?

Line 143: What was the rhythm of repetitions?

Line 146: Did you assess active ROM to determine range? Or was it visually assessed only?

Line 154: Explain more about the pain and how you knew it was not related to participation in the project.

Line 203: Why 25? Did the previous study use similar statistics? Could you calculate effect size?

Lines 251-305: This reads a lot like an introductory exercise physiology text. The capillary discussion seems out of place because it has nothing to do with your results. A lot of what you write about can be connected to muscle fiber type. Consider adding information about the muscle fiber type of the muscles that you tested. Connect the different ideas better and make a stronger conclusion so that it is clear why it is included in the discussion.

Line 296: Insert reference

Line 298: Not sure how you came up with this conclusion. Based on what you have written here, I would disagree. Please add more specifics.

Line 301: The plastic nature of human muscle just doesn’t change that easily. Muscle fiber types don’t change from a type I to a type IIX because of the demands placed upon it.

Line 321: You may want to give some of this information earlier in the article as the reason why you didn’t compare your results to an isokinetic machine.

Line 345 & Practical applications: Seems like this information would be great to have in the introduction, especially because you don’t specifically discuss your results and how others can use your results as a practicing clinician.

Author Response

Reviewer 1 comment:

The article has merit and practical use. The descriptions of the exercise are clear and the pictures help. There are other articles and organizations that have evaluated sub-maximal testing to predict 1 RM, a discussion of this would add to the article. The discussion talks more about the physiology than what this means for other clinicians. Please work on describing the “so what?” to the article. How can people use this information moving forward?

OUR REPLY: We agree that the article has merit and practical use. We have now emphasized the practical application of the results in the introduction section. Which we feel also is in accordance with the last comment from this reviewer, regarding the practical applications of the results.

In the discussion we attempt to provide the underlying physiological mechanisms of our results to the reader, as to why it was necessary to have two different formulas, one for the upper body muscles and one for lower body muscles. We have, also in accordance with other comments, adjusted the Discussion accordingly.    

Reviewer 1 comment:

Line 109: Are there any references for why you stretched the bands 50-100 times? Is there a point where the bands loose tensile strength/elastic damage?

OUR REPLY: This was done according to the reference of Andersen et al 2016, SJMSS (https://pubmed.ncbi.nlm.ih.gov/27185407 ). This reference has now been added to the methods. We are unaware at what threshold the bands loose tensile strength, but according to our own experience they can be used for one to two years. As we used new elastic bands we don’t think this affected the study results.  

Reviewer 1 comment:

Line 143: What was the rhythm of repetitions?

OUR REPLY: The rhythm of repetitions was 1,5-2 seconds in the concentric phase and 1-1,5 seconds in the eccentric phase). This has information has been added to the methods section.  

Reviewer 1 comment:

Line 146: Did you assess active ROM to determine range? Or was it visually assessed only?

OUR REPLY: This was visually assessed. We have now added this to the Methods.

Reviewer 1 comment:

Line 154: Explain more about the pain and how you knew it was not related to participation in the project.

OUR REPLY: This was according to the participants own information about their history of pain and injuries. We have now added this to the Methods.

Reviewer 1 comment:

Line 203: Why 25? Did the previous study use similar statistics? Could you calculate effect size?

OUR REPLY: The previous study did not use exactly the same statistics, but based on the known inverse relationship between intensity and repetitions we could have performed the study with fewer participants. However, we felt that 9 subjects were too few to make any generalizations. Thus, we wanted to test as many as was practically possible. Within the time frame of the study, we estimated that we could test between 20-30 participants.

Reviewer 1 comment:

Lines 251-305: This reads a lot like an introductory exercise physiology text. The capillary discussion seems out of place because it has nothing to do with your results. A lot of what you write about can be connected to muscle fiber type. Consider adding information about the muscle fiber type of the muscles that you tested. Connect the different ideas better and make a stronger conclusion so that it is clear why it is included in the discussion.

OUR REPLY: We agree that the capillary discussion is rather unclear in regard to our results and we have opted for removed this section from the Discussion.

Reviewer 1 comment:

Line 296: Insert reference

OUR REPLY: A reference to the article in question is inserted at the end of the sentence.

Reviewer 1 comment:

Line 298: Not sure how you came up with this conclusion. Based on what you have written here, I would disagree. Please add more specifics.

OUR REPLY: Our point here is that it appears that the muscles in the lower extremities sees to have a greater oxidative capacity, and we added our own physiological reflections on the reason for this difference, namely on the training stimulus. We have now removed that from the discussion.

Reviewer 1 comment:

Line 301: The plastic nature of human muscle just doesn’t change that easily. Muscle fiber types don’t change from a type I to a type IIX because of the demands placed upon it.

OUR REPLY: We agree with the reviewer that the nature of human muscle fibers do not change easily, and they certainly do not change fiber type. We only point out that human muscle fibers can adapt to the demands that is placed upon them, by changing their metabolism.

Reviewer 1 comment:

Line 321: You may want to give some of this information earlier in the article as the reason why you didn’t compare your results to an isokinetic machine.

OUR REPLY: This is a good point, and we have now addressed this in the Method section. 

Reviewer 1 comment:

Line 345 & Practical applications: Seems like this information would be great to have in the introduction, especially because you don’t specifically discuss your results and how others can use your results as a practicing clinician.

OUR REPLY: This is a good point, we have now incorporated these aspects into the Introduction section. 

Reviewer 2 Report

This article presents the results assessing muscular strength in healthy individuals with the use of elastic bands.

The article is clearly presented, however, few questions were raised.

  1. What was the type of anti-inflammatory non-steroidal anti-inflammatory drugs or steroidal drugs?
  2. Why the age was so wide? The authors should explain why they choose to include people raging 18-67 years.
  3. In table 1 smokers/non-smokers were 1/26 what gives 27, however, the total number of participants was 26.
  4. There were any differences in maximal strength between woman and man or did the maximal strength correlated with age?

Author Response

Reviewer 2 comment:

This article presents the results assessing muscular strength in healthy individuals with the use of elastic bands.

The article is clearly presented, however, few questions were raised.

    What was the type of anti-inflammatory non-steroidal anti-inflammatory drugs or steroidal drugs?

OUR REPLY: One of the exclusion criteria was use of anti-inflammatory drugs. We did not specify what type of drugs could fall under this, but it is over-the-counter pain relief drugs and prescription pain relief drugs of any sort, or in general.    

Reviewer 2 comment:

    Why the age was so wide? The authors should explain why they choose to include people raging 18-67 years.

OUR REPLY: The wide age range of inclusion was to make the findings more generalizable, although the majority were in the lower range of 18-67 years.

Reviewer 2 comment:

    In table 1 smokers/non-smokers were 1/26 what gives 27, however, the total number of participants was 26.

OUR REPLY: We meant 1 out of 26, but we can see that it should be 1/25 to be clearer. We have corrected this in the table.

Reviewer 2 comment:

    There were any differences in maximal strength between woman and man or did the maximal strength correlated with age?

OUR REPLY: Yes, the men were generally stronger than the women, but the relation between repetitions and intensity was not significantly different between men and women. We have now added a sentence about this to the Results. Regarding age, the sample size was too small to test the influence of age, as the majority of participants were in the lower age range.

Reviewer 3 Report

Nicely considered and conducted project on a meaningful, and useful topic. Authors are to be commended for their thoroughness and for a fine, practical contribution to the field. Only concern is that they may want to be a bit more cautious with speculative comments made on lines 270-275 as no muscle biopsies were taken here so there is no direct evidence regarding metabolic capacity, capillarization between lower, and upper body muscles can be made here.

Author Response

Reviewer 3 comment:

Nicely considered and conducted project on a meaningful, and useful topic. Authors are to be commended for their thoroughness and for a fine, practical contribution to the field. Only concern is that they may want to be a bit more cautious with speculative comments made on lines 270-275 as no muscle biopsies were taken here so there is no direct evidence regarding metabolic capacity, capillarization between lower, and upper body muscles can be made here.

OUR REPLY: True. We have now removed this part.